# Keeping Cats Safe at Home (KCSAH): Lessons Learned from a Human Behaviour Change Campaign to Reduce the Impacts of Free-Roaming Domestic Cats

**DOI:** 10.3390/ani15243554

**Published:** 2025-12-10

**Authors:** Gemma C. Ma, M. Carolyn Gates, Katherine E. Littlewood, Sarah Zito, Brooke P. A. Kennedy

**Affiliations:** 1The Royal Society for the Prevention of Cruelty to Animals New South Wales, Yagoona, NSW 2199, Australia; 2School of Veterinary Science, The University of Sydney, Camperdown, NSW 2050, Australia; 3AkoVet Limited, Palmerston North 4410, New Zealand; carolyn.gates@akovet.org (M.C.G.); kat.littlewood@akovet.org (K.E.L.); 4RSPCA Australia, Deakin, ACT 2600, Australia; szito@rspca.org.au; 5School of Environmental and Rural Science, University of New England, Armidale, NSW 2350, Australia; bkenne27@une.edu.au

**Keywords:** cats, roaming, containment, desexing, behaviour change, community engagement, social media, animal welfare, predation

## Abstract

Domestic cats are valued companions for many people, yet their tendency to roam can create challenges for biodiversity, communities, and the cats themselves. Encouraging owners to keep cats contained is increasingly recognised as a priority in Australia but requires strategies that address the practical and behavioural barriers that people face. The Keeping Cats Safe at Home project was a four-year initiative led by the Royal Society for the Prevention of Cruelty to Animals New South Wales (RSPCA NSW) in partnership with 11 councils from 2021 to 2024. The project tested a suite of approaches, including social marketing campaigns, school education, community events, and subsidised desexing and microchipping, to encourage safer and more sustainable ways of caring for cats. These activities achieved a broad reach through digital platforms and direct engagement, and resulted in thousands of cats being desexed and microchipped. Ecological monitoring indicated reductions in roaming cat numbers in several areas, and many participating councils recorded fewer nuisance complaints and impoundments. Owners who encountered campaign messages also reported feeling more able and motivated to keep their cats contained, although measuring actual behaviour change proved difficult. The project demonstrates that large-scale behaviour change programmes can improve outcomes for cats, communities, and wildlife, but highlights the need for sustained investment, stronger veterinary capacity, and consistent monitoring to support long-term success.

## 1. Introduction

Domestic cat management is a complex but increasingly topical area of focus for governments, animal welfare organisations, conservation groups, and communities across Australia [1]. This is because domestic cats are highly valued as companions and integral members of many households; however, they can also negatively impact wildlife, ecosystems, biodiversity, and human communities, particularly when they are allowed to roam freely [2]. Research suggests that domestic cats kill large numbers of native animals each year in Australia, with predation pressures greatest in urban and peri-urban areas where owned cat densities are highest [3]. While several studies have shown that measures such as using collars fitted with anti-predation devices [4,5,6,7] or incorporating different feeding programmes or structured play into domestic cat care routines [8] can help to reduce predation rates, they do not completely eliminate hunting, roaming, and other perceived nuisance behaviours [9,10,11,12].

As such, there has been growing interest in promoting cat containment as a practical strategy to mitigate risks to biodiversity, communities, and the cats themselves [1]. Containment encompasses a spectrum of approaches, from dusk-to-dawn curfews through to 24 h confinement to the owner’s property using cat-proof fencing, enclosures, or indoor-only housing [13]. It may also involve cat exclusion zones in sensitive habitats [14], boundary deterrent devices [15], or supervised outdoor access such as leash-walking [16]. Given the known risks and behaviours associated with roaming [17,18], containment should theoretically reduce predation, nuisance behaviours, and unplanned litters in certain settings, while also lowering risks of injury, poisoning, and disease transmission for the cats themselves. Some owners have also reported greater peace of mind and stronger relationships with their animals when cats are safely contained [19]. Contained environments must provide engagement opportunities that support positive affective experiences and a healthy weight; without these, risks of frustration, anxiety, inactivity, undesirable behaviour, and poor welfare rise [20,21,22]. Ensuring opportunities for predatory play, exploration, and choice through indoor resources, cat-proof gardens, safe outdoor enclosures, and consistent positive human interaction is critical to making containment effective for both biodiversity and feline welfare [19,23,24].

According to the 2025 Pets in Australia report [25], cat ownership in Australia has risen from 27% of households in 2019 to 34% in 2025, equating to an estimated 5.8 million owned cats, or an average of 1.6 cats per cat-owning household. While there has been a noticeable shift toward more contained lifestyles with the proportion of indoor-only cats increasing from 36% in 2019 to 48% in 2025, the uptake of other practices has remained relatively stable or declined, with desexing rates currently sitting at 84%, microchipping at 78%, and vaccination at 66% [25]. These national trends, which highlight potential gaps in how some owners currently care for their cats, are reflected in local situational analyses [26], particularly in communities with high shelter admission rates. Rand et al. [27] found that while overall desexing rates among owned cats were high (91%), more than one-quarter of cats aged 4–11 months (26%) remained undesexed, placing them at high risk of contributing to accidental litters. Containment was also inconsistent, with only 51% of owners keeping their cats fully contained and 15% allowing free roaming, and a further 33% containing their cats only some of the time [27]. Similar patterns were reported by Dutton-Regester and Rand [28] in households enrolling undesexed cats in a free sterilisation programme, where cost was a major barrier to desexing, but awareness of optimal feeding, housing, and preventive health was also low, with only 15% of cats having visited a veterinarian and 28% being vaccinated. Both studies highlight the gap between owner intentions and actual management practices, with socio-economic constraints, limited awareness of best-practice care, and cultural norms influencing behaviour.

Local councils in Australia are responsible for domestic cat management under state-based legislation. In practice, however, the levels of action vary substantially, with some councils undertaking limited or no cat management activities [29]. While public surveys show strong in-principal support for containment, actual compliance lags behind. In a large Australian survey of 6808 respondents, Elliott et al. [30] found that 88.6% of cat owners agreed cats should be contained at night, yet only 76.9% reported doing so. Support for 24 h containment was more divided: 47.3% of owners agreed it should be the norm, which corresponds with the 46.5% who also reported keeping their cats fully contained. Non-owners were significantly more likely than owners to support 24 h containment, suggesting higher public tolerance for stricter restrictions but also highlighting tensions between community expectations and owner practices. Local government data reinforce this picture: almost a third of councils now mandate curfews, 24 h containment, or cat exclusion zones, particularly in Victoria, South Australia, and the Australian Capital Territory (ACT). However, compliance monitoring remains limited, and enforcement is inconsistent. Concerns have also been raised that strict legal mandates, if introduced without adequate support, could increase abandonment or surrender rates, as well as compromise cat welfare if they are not contained in appropriate environments [31].

As a result, there has been an increasing reliance on the voluntary uptake of cat containment practices, yet barriers remain, including cost, housing design, and cultural beliefs that roaming is a natural behaviour for cats [32,33,34]. Furthermore, the presence of semi-owned cats (i.e., cats receiving some care from a person or people but without full ownership responsibility) is common, blurring the lines between owned and unowned populations and limiting accountability for care [35,36,37]. Case studies also show that even when owners express strong intentions to contain their cats, these do not always translate into behaviour once practical challenges emerge [38]. Together, these findings suggest that while containment is gaining traction as a management tool, achieving widespread adoption will require carefully designed strategies that bridge the gap between intention and practice, combining community engagement, education, incentives, and support for owners.

To address these challenges, RSPCA NSW secured an AUD 2.5 million grant from the NSW Government through its Environmental Trust to deliver a four-year programme exploring the application of different human behaviour change strategies to domestic cat management. Implemented across 11 local councils in NSW from 2021 to 2024, the *Keeping Cats Safe at Home (KCSAH)* project aimed to reduce both the size and ecological impacts of free-roaming domestic cat populations by encouraging owners to voluntarily contain their cats. Drawing on behaviour change research as well as extensive feedback from consultation with local stakeholders, the KSCAH project developed and implemented a suite of interventions, including marketing campaigns, school-based education, community events, and training resources to support veterinarians, rescue organisations, and local councils in engaging owners in conversations about cat containment. To further reduce barriers to care that enable containment practices, the project also offered free desexing and microchipping targeted at priority populations of cat caregivers.

The objective of this manuscript was to provide a comprehensive overview of the logistics involved in designing, implementing, and evaluating the KCSAH project, offering practical guidance for other jurisdictions considering similar programmes.

## 2. Project Overview

The KCSAH project was delivered in a staged approach (Figure 1). The **project initiation phase** (January to June 2021) focused on recruiting the governance team, expert advisors, and participating councils. The **stakeholder engagement phase** (June to November 2021) gathered insights into the current domestic management landscape to inform programme design, monitoring, and evaluation. The **campaign development phase** (January 2022 to March 2023) involved designing messaging and resources, followed by the **campaign implementation phase** (July 2022 to December 2024), during which project activities were rolled out across councils. Concurrently, the **evaluation phase** (June to November 2024) involved collecting and analysing data to assess the project’s impacts.

### 2.1. Project Initiation Phase (January to June 2021)

The KCSAH project commenced on 1 January 2021 with the establishment of a multi-stakeholder advisory group, recruitment of a project manager, and engagement of expert consultants in behaviour change and animal ecology to support the design of project resources and the monitoring and evaluation framework.

A key early activity was the recruitment of local councils to participate as trial regions. Councils were selected because they represent the jurisdictional unit with statutory responsibility for companion animal management under the *Companion Animals Act 1998 (NSW)*. Recruitment aimed to secure a cohort of councils that reflected a diverse demographic and geographic profile, enabling evaluation of project effectiveness across different contexts.

RSPCA NSW, in consultation with the advisory group, established a protocol for inviting expressions of interest (EOIs) and developed criteria for council selection. EOIs were advertised and invited from all 128 councils in NSW between 22 January 2021 and 5 March 2021. A total of 29 EOIs were received, of which 11 councils were selected as project partners. Selection prioritised councils that demonstrated commitment to project objectives, wildlife protection, and humane, evidence-based cat management; proximity to sensitive wildlife habitats; capacity for strong community engagement; and willingness to desex all cats prior to rehoming from council pounds. Additional considerations included the feasibility of project delivery under COVID-19 restrictions and a commitment to provide relevant data for monitoring and evaluation.

Table 1 summarises the demographic characteristics of the participating councils, and their geographic distribution across New South Wales is shown in Figure 2.

### 2.2. Stakeholder Engagement Phase (June to November 2021)

A central component of the KCSAH project was engagement with key stakeholder groups to assess current attitudes, practices, and barriers related to cat containment. Stakeholders included cat owners, semi-owners and caretakers, local councils, veterinarians, animal rescue organisations and shelters, and wildlife organisations. A consultation process was undertaken through structured interviews, workshops, and surveys to ensure that diverse perspectives were incorporated into the campaign design.

The social science research component focused particularly on cat caregivers, with two strands of work published as peer-reviewed manuscripts. One study characterised the practices and motivations of cat semi-owners, who care for cats without formally recognising ownership [36]. A second study investigated the broader factors influencing containment decisions among cat caregivers using the Capability–Opportunity–Motivation Behaviour (COM-B) framework to identify key drivers and barriers [39] and to segment cat owners into six distinct profiles that could be used to target campaign messaging [40]. Profile labels, criteria, and exemplar messages are provided in Appendix A.

Psychological capability was identified as the most significant barrier across both studies. Caregivers who believed it would be too difficult to prevent roaming, lacked confidence in their ability to restrict their cat’s movements, or were unsure how to provide opportunities for their cats to experience positive welfare states in a contained environment were far more likely to allow roaming. The six profiles underscored the importance of tailoring interventions: Profiles 1–2, who viewed roaming as beneficial and showed little concern about risks, required strategies that increased motivation and directly addressed beliefs about cat welfare; Profiles 3–4, who lacked confidence and understanding of how to contain their cats effectively, needed skills and support to build psychological capability; while Profiles 5–6, who showed strong alignment with COM-B themes and high community motivation, were best supported with tools and reinforcement to consolidate containment intentions.

### 2.3. Campaign Development Phase (January 2022 to March 2023) and Implementation Phase (July 2022 to December 2024)

Insights from the stakeholder engagement guided the design of a portfolio of resources and activities that aimed to build capability, opportunity, and motivation for human behaviour change while normalising containment as part of routine cat care.

#### 2.3.1. Website and Information Resources

A dedicated Keeping Cats Safe at Home project landing page was embedded within the RSPCA NSW website and launched in mid-2022, serving as the central platform for campaign messaging and resources. The site was updated regularly throughout the project to ensure its relevance and to address emerging issues identified during stakeholder engagement.

A suite of high-quality information resources was developed in collaboration with veterinarians and clinical behaviourists. These included brochures, fact sheets, and adoption-pack inserts that provided practical guidance on cat behaviour and communication, safe housing, and strategies for transitioning roaming cats to an indoor or contained lifestyle. Where appropriate, materials were also translated into languages commonly spoken in the target areas to improve accessibility.

Distribution channels were deliberately chosen to maximise trust and uptake. Information resources were provided directly to veterinary clinics, councils, and rehoming organisations so that they could be offered during routine interactions with cat caregivers, including veterinary consultations and the adoption process.

All education resources were also made freely accessible online via the project landing page and optimised for search engines to capture caregivers seeking information about cat care and management. Materials were designed for flexible use, suitable for reading online, downloading, or printing at home, thereby reducing barriers to access.

#### 2.3.2. Marketing Campaign

The KCSAH marketing campaign aimed to increase the psychological capability and motivation of cat caregivers to contain their cats, using human behaviour change techniques such as demonstration of the behaviour, social support, goal setting, and action planning [39,41]. Campaign activities were organised around four main delivery streams:

##### Social Media Posting

Quarterly themed bursts of content were shared via RSPCA NSW’s social media channels (Facebook and Instagram), beginning in June 2022. Content included recovery stories of cats who were injured while roaming and then successfully adapted to containment, competitions, activity and resource tips, and examples of containment systems. Informational posts were interspersed with highly engaging pieces to sustain interest. Partner councils, veterinary practices, and rehoming organisations were encouraged to share or adapt campaign content. The campaign also benefited from strong media interest, with additional radio, television, and print coverage helping to extend the reach of its messaging.

##### Email Newsletter

A triggered twelve-edition email newsletter was developed, delivering monthly content to subscribed cat caregivers over a one-year period. Each edition combined practical cat care advice with embedded messages about the benefits and feasibility of containment. To encourage initial subscriptions, the newsletter was promoted through a cat grass giveaway, linking an appealing incentive with the campaign’s educational content.

##### Blog Articles

A series of blog articles was published through the campaign website, providing accessible, story-driven content for cat caregivers. The 22 articles combined educational pieces on cat care and management with updates on project activities, positioning the campaign website as both an information hub and a narrative record of the project.

##### Paid Advertising Campaign

In 2024, KCSAH collaborated with a social marketing agency to develop a paid advertising campaign titled *Not All Cat Videos Are Funny*. Two creative concepts were tested through focus groups with cat caregivers whose cats roam, ensuring message alignment with the target audience. The final campaign combined video storytelling with digital promotion across catch-up television, YouTube, Facebook, and Instagram, aiming to increase awareness and motivation to contain cats among a broader audience than could be reached through organic channels alone.

#### 2.3.3. Partner Packages

To extend the campaign’s reach through trusted intermediaries, *Keeping Cats Safe at Home* developed comprehensive partner packages for councils, veterinary practices, and rehoming organisations. These packages included printed brochures, fact sheets, and collateral tailored to support staff in initiating conversations with cat caregivers about the risks of roaming, strategies for containment, and how to provide cats with the necessary care at home. By equipping local professionals with consistent and evidence-based materials, the project sought to embed campaign messages within everyday interactions between caregivers and organisations they already viewed as reliable sources of advice [30].

Partner packages were distributed to all 11 participating councils, over 60 veterinary practices, and more than a dozen rehoming organisations. In addition, the education resources were incorporated into RSPCA NSW adoption packs for cats and kittens across the state and made available through the organisation’s three veterinary clinics. This integration attempted to ensure that cat caregivers encountered containment messaging at multiple points of contact, including during adoptions, veterinary visits, and routine engagement with councils, normalising containment as an ownership behaviour and reinforcing campaign objectives.

#### 2.3.4. School Education Packages

Recognising the role of early education in shaping long-term attitudes toward animal care and wildlife protection, KCSAH developed a curriculum-linked education package for primary school students aged 5–12 years. The package included a lesson plan aligned with the NSW syllabus, a supporting PowerPoint presentation with embedded educational videos, and supplementary classroom activities. These materials were designed for delivery by school teachers or trained volunteers, providing age-appropriate lessons on cat care and management, the risks associated with roaming, and the impacts of cat predation on native wildlife.

To reinforce classroom learning, the education package was complemented by a *Keeping Cats Safe at Home* magazine, an illustrated activity booklet, and a take-home poster, encouraging children to continue the conversation about cat care within their households. All resources emphasised the dual message of valuing cats as companions while recognising their potential impacts on wildlife, thereby normalising containment behaviours from a young age. All educational resources were also made freely available for download through the project website.

The programme also included creative engagement tools. A children’s book, *Stay Safe Clancy*, was developed in collaboration with Northern Beaches Council and a local artist. The book was distributed to local libraries, primary schools, and community centres across participating councils, broadening reach beyond the classroom and embedding campaign messages in family and community settings.

#### 2.3.5. Community Events and Engagement

Direct engagement with the public and professional stakeholders formed a core component of the KCSAH campaign. The project team participated in 34 community and industry events throughout the project’s life, reaching diverse audiences, including cat caregivers, veterinarians, and allied animal care professionals.

High-profile events such as the *Cat Lovers Festival* (2023 and 2024) and the *Vet Expo* (2022 and 2024) provided opportunities to interact with large audiences of cat enthusiasts and veterinary professionals. These events enabled the team to showcase containment solutions, distribute resources, and hold face-to-face conversations about cat care and the risks of roaming.

In addition, the KCSAH team collaborated with partner councils to participate in local events across each of the 11 participating local government areas. Agricultural shows, Family Fun Days, and Healthy Pet Days created opportunities to embed campaign messages within community contexts and reach cat caregivers who may not engage through digital or formal education channels.

#### 2.3.6. Targeted Desexing Programme

Desexing is critical for reducing cat roaming and overpopulation, yet cost remains a major barrier for many caregivers. To address this, free desexing and microchipping programmes were established in most of the 11 KCSAH partner council areas, with more intensive “StrayCare” initiatives implemented in six councils to improve uptake among semi-owned and unowned cat groups that contribute disproportionately to new litters. The decision of whether and how to implement a cat desexing programme and how much of the project budget would be allocated to desexing was based on discussions with each partner council based on the specific cat-related challenges in their area. These programmes were delivered in collaboration with 26 local veterinary practices, council animal management teams, and local cat rescue volunteers, creating strong local partnerships. A key focus was on supporting semi-owners to take long-term responsibility for the cats in their care by removing financial and logistical barriers to desexing.

Eligibility for the programmes was deliberately broad, limited only by geographic area (whole local government area, postcode, or selected suburbs, depending on population). Within this, targeting was achieved by prioritising enrolment of caregivers of ‘stray’ cats and people caring for multiple cats, regardless of whether caregivers considered them owned or unowned. Promotion was intentionally restricted and highly localised, using channels such as direct referrals from council animal management workers and rescue groups, as well as letterbox drops and door-to-door outreach in areas with known unowned cat populations.

### 2.4. Monitoring and Evaluation Framework

A comprehensive monitoring and evaluation framework was established to track project delivery, measure outcomes, and identify areas for improvement. The framework combined quantitative indicators of reach, ecological change, community impact, and caregiver behaviour with qualitative feedback from staff and participants, ensuring both accountability and opportunities for learning.

#### 2.4.1. Project Reach

Project reach was assessed through digital analytics, event participation records, and stakeholder engagement tracking. Metrics included the number of website visits, blog/article views, social media reach, downloads of educational resources, direct participation in subsidised desexing and microchipping programmes, attendance at community events, uptake of school education resources, and the level of veterinary and community organisation involvement. These indicators provided a quantitative measure of campaign penetration across participating councils.

#### 2.4.2. Ecological Impacts

Ecological impacts were evaluated using camera traps and distance sampling transect drives across four councils. Distance sampling transects were conducted to estimate the population size of free-roaming cats. At the same time, camera traps were deployed to determine the spatial and temporal movements of free-roaming cats and any overlap with wildlife. Two council areas were in Greater Sydney [42] and two were on the North Coast of NSW (unpublished). Additional ecological insights were drawn from a complementary research project examining cat movement patterns and camera-trap detections of cats and other wildlife at Sydney’s North Head [43]. Together, these datasets allowed for the assessment of changes in roaming cat density and patterns of wildlife presence in the study area.

#### 2.4.3. Community Impacts

Community-level impacts were monitored using routinely collected council data. Annual rates of cat-related nuisance complaints and cat impoundments were compared to pre-intervention baselines (four-year averages) to assess whether project activities were associated with reductions in free-roaming cat problems.

#### 2.4.4. Human Behaviour Change Impacts

At the start of the project, a statewide online cross-sectional survey was conducted as part of the target audience consultation to collect information from cat owners and semi-owners about their perspectives on free-roaming cats and current cat ownership behaviours, with findings reported in Ma et al. [36] and Ma and McLeod [40]. At the conclusion of the three-year project, a second statewide panel-based survey of cat owners was undertaken in November 2024. The purpose of this follow-up survey was to assess the effectiveness of the KCSAH campaigns by capturing data on current cat ownership behaviours, attitudes towards free-roaming cats, and awareness of campaign messaging.

## 3. Project Outcomes

### 3.1. Project Expenditures

The KCSAH project was delivered over four years with a total budget of AUD 2.52 million. Table 2 provides a breakdown of expenditures by financial year and activity area. Salaries and consultancy (38%), the targeted desexing programme (32.9%), and the social media marketing campaign (17.3%) accounted for the majority of project costs. It was not possible to determine the precise allocation of full-time equivalent (FTE) effort by project staff across individual behaviour change campaign activities, as roles were fluid and responsibilities overlapped. The project also leveraged existing RSPCA infrastructure and in-kind support (e.g., legal, finance, communications, outreach), which enabled delivery at a greater scale than would have been possible within the direct budget allocation.

### 3.2. Project Reach

The KCSAH project achieved engagement that exceeded initial expectations across digital, community, education, and professional channels (Table 3). The campaign landing page attracted over 87,000 visits, complemented by strong readership of blog articles and downloads of educational resources. Social media activities reached more than 3.5 million people across multiple campaign waves, while direct participation included over 4000 caregivers in desexing, microchipping, and incentive programmes. In-person outreach resulted in more than 36,000 community members engaged at events and over 1400 students participating in school programmes. Veterinary and community partnerships were also extensive, with all local veterinarians involved in some form and 31 additional organisations supporting project delivery.

### 3.3. Ecological Outcomes

Distance sampling transect drives indicated reductions in the estimated free-roaming cat populations across three participating local government areas, with declines of 51% in the Blue Mountains, 35% in Campbelltown, and 25% in Tweed Shire (Figure 3; Table 4). The small sample size (*n* = 3) precluded further analysis to assess the statistical significance of the change in cat population sizes, but it provides preliminary evidence of cat population reduction between these two time points. Camera traps showed that the cats still roaming were doing so at the same times and locations as before. The cats were also still overlapping with wildlife movements and therefore having the same potential impacts (both disturbance and predation).

### 3.4. Community Outcomes

Community-level indicators showed substantial variation across participating councils (Table 5). Several areas achieved marked reductions in both nuisance complaints and cat impoundments, suggesting positive impacts of project activities on local cat management. In Campbelltown, Parramatta, Hornsby, and the Northern Beaches, nuisance complaints fell by 47–64%. Conversely, complaint volumes rose in some regions, including the Blue Mountains and Byron Shire, reflecting differences in local reporting systems and community engagement.

Trends in impoundments were similarly mixed. Significant decreases were observed in the Blue Mountains (−54%), Campbelltown (−59%), Parramatta (−73%), and Walgett (−100%), while increases occurred in Northern Beaches (+53%) and Weddin (+380%). The rise in Weddin was attributed to increased trust in council staff, who were perceived as more approachable and focused on providing assistance rather than punitive actions, which encouraged greater reporting and handover of cats. In contrast, Northern Beaches did not implement targeted desexing programmes during the project.

### 3.5. Human Behaviour Change Outcomes

Overall, human behaviour change indicators demonstrated some positive shifts in cat guardianship. The post-intervention survey included 2039 cat owners from across NSW. Results showed that 46% of respondents reported 24 h containment, 26% practiced night curfews, and 28% allowed their cats to roam freely. This represented a lower proportion of cats reported as fully contained compared with the baseline survey in 2021, where 65% of owners indicated 24 h containment, 24% practiced a night curfew, and 11% allowed free roaming [40] (Ma & McLeod, 2023). This apparent decline is unlikely to reflect a true statewide change and more plausibly reflects differences in recruitment channels and respondent profiles between the two surveys. Owners in the 2024 survey who recalled exposure to KCSAH campaign messaging reported significantly higher levels of capability, opportunity, and motivation to contain their cats than those who had not encountered the campaign. However, the survey response rates from the participating councils were too low to fully evaluate the impacts of the campaign on promoting behaviour change.

Other human behaviour change outcomes were more directly measurable (Table 6). Over 2700 cats were desexed through the project, including 877 before the age of six months, substantially exceeding the target for pre-pubertal desexing. More than 17,000 new cats were registered across participating councils, with proportional increases ranging from modest gains in Shoalhaven (+2%) to large rises in Weddin (+75%). Microchipping uptake was also strong, with 1721 cats microchipped through the project. By contrast, uptake of discounted cat enclosures was low (31 purchases against a target of 300), which may reflect not only caregiver preference for alternative containment approaches such as indoor-only living, modified fencing, or supervised outdoor access but also ongoing cost and practical barriers—particularly for renters unable to install enclosures or for households concerned about aesthetics or space.

### 3.6. Lessons Learned

The implementation and evaluation of the KCSAH project highlighted several systemic, programmatic, and contextual factors that shaped outcomes and will be critical to consider in future initiatives.

#### 3.6.1. Regulatory Context as a Limiting Factor

The most significant barrier identified was the regulatory framework under the *Companion Animals Act 1998* (NSW). Ambiguities in the Act—including the absence of clear definitions for unowned or “stray” cats, lack of clarity around councils’ responsibilities, and limited enforcement options for nuisance cats—restricted the effectiveness of both the marketing campaign and the targeted desexing programme. Cost barriers, such as registration fees and annual permit fees for undesexed cats, also acted as disincentives for community members to assume ownership of unowned cats they were already caring for.

#### 3.6.2. Segmented Audiences and Human Behaviour Change Capability

The project confirmed that cat caregivers who allow their cats to roam are not a homogeneous group. Their capability, opportunity, and motivation to contain cats vary considerably. Some expressed psychological barriers, believing they could not prevent roaming or that containment would compromise their cat’s welfare. Households that acquired cats passively (“adopted by circumstance”) were more likely to allow their cats to roam, highlighting the need for targeted support. Rural and semi-rural owners, including those with “working cats,” required different messaging and incentives compared with suburban pet owners. These findings reinforce the need for tailored strategies rather than one-size-fits-all messaging.

#### 3.6.3. Programmatic and Contextual Challenges

External events, including COVID-19 restrictions and severe flooding, disrupted early project activities, delaying community engagement and service delivery. The migration of unowned cats into project areas, particularly in rural councils, undermines containment and desexing gains, emphasising the importance of sustained, regionally coordinated programmes addressing both owned and unowned populations.

#### 3.6.4. Optimising Communication Strategies

The social media marketing campaign yielded several lessons for maximising reach and engagement. Staggered bursts of content helped reduce audience fatigue, while low-cost, do-it-yourself indoor cat activity videos resonated strongly. Storytelling through cautionary tales of cats injured while roaming, followed by successful containment, proved especially engaging. Interactive competitions that invited cat caregivers to share photos, videos, and stories generated high participation, suggesting that peer-to-peer sharing is a powerful motivator for engagement.

#### 3.6.5. Barriers to Desexing Participation

Uptake of subsidised desexing varied across councils. Target audiences, such as semi-owners and multi-cat caregivers, often faced overlapping barriers, including social isolation, a lack of transportation, difficulties in handling cats, and longstanding mistrust of local government. Programme success improved when participation requirements, such as mandatory microchipping, were minimised and assurances were provided that enforcement would not follow if cats were unregistered or unmicrochipped, or if their caregivers might otherwise face compliance penalties, impound fees, or fines. Flexibility in the number of cats enrolled per caregiver, as well as efforts to build relationships and trust, further boosted engagement.

#### 3.6.6. Veterinary Workforce Capacity

Persistent veterinary workforce shortages constrained programme delivery throughout the project’s lifespan. These shortages limited clinic participation and inflated desexing costs (~AUD 300 per surgery performed), reducing programme efficiency. This structural challenge has broader implications for the scalability of companion animal management programmes in Australia.

## 4. Discussion

The KCSAH project provides one of the most comprehensive examples to date of a community-driven human behaviour change programme for domestic cats in Australia. The findings highlight both the opportunities and the challenges of implementing large-scale, multi-channel interventions in this space.

The first key insight was the importance of developing tailored approaches for targeting semi-owners. This group is critical for domestic cat management programmes, given that the population of semi-owned and unowned domestic cats in Australia is estimated to be anywhere from 0.7 to 2 million, or approximately 60–100 cats per 1000 human residents [44,45,46]. In the KCSAH study sample, 7% of residents reported engaging in some form of semi-ownership [36], which is lower than the 22% of residents who reported feeding stray cats in a Victorian study nearly two decades earlier [35]. This difference may reflect geographic variation, but it is also possible that shifting messaging and interventions around stray cat management have influenced behaviour over time.

While many semi-owners also have cats of their own, they are often unwilling or unable to formally adopt additional stray cats. Reported reasons include not wanting another animal, concerns about impacts on resident cats, or restrictions associated with their accommodation [37,47]. Furthermore, many value the perceived independence of cats [35], making full containment an unlikely outcome for this group. However, it is important to note that some of these findings are from a study focused on individuals surrendering stray cats to shelters [46], a subset of semi-owners who have the capacity and willingness to capture and transport cats, which may not accurately represent the broader population of semi-owners. Regardless, management strategies should take a human behaviour-change approach and (1) encourage the community to notice and act to ensure unowned cats are actively and appropriately managed and (2) engage with semi-owners to create a long-term plan for each cat, including designating a caregiver, or ‘owner’, responsible for their ongoing care.

For semi-owners, the requirement to register cats and pay the associated lifetime fee of approximately AUD 70 was a major deterrent to microchipping during the KCSAH targeted desexing campaign, even though the microchipping was free. Some semi-owners also expressed concerns about being held legally responsible if the cat became sick or injured and required costly treatment, or if it was declared a nuisance. Under current NSW law, anyone providing care for a cat can be deemed its legal owner with all the associated responsibilities, regardless of whether the cat is microchipped and registered. In practice, however, the regulations for nuisance cat behaviour in NSW are weak; while cats can technically be declared a “nuisance,” the threshold for proving an offence is sufficiently high that this almost never occurs. This experience highlighted that financial incentives alone are not enough and that long-term change depends on engaging with semi-owners while also addressing the underlying sources of unowned cat populations. In Australia, the law generally treats anyone providing ongoing care as the legal “owner,” even if they do not view themselves that way, which can create reluctance to formalise the relationship. Other countries have established formal colony caretaker systems (for example, Italy (Law 281/1991), Greece (Law 4039/2012 and amendments), Spain (National Animal Welfare Law 7/2023), and Brazil (municipal frameworks such as São Paulo Decree 48.533/2007 and Rio de Janeiro Law 4.877/2008)), which authorise recognised community caretakers to feed and monitor free-roaming cats without assuming full legal liability. While such models reduce barriers, they also risk normalising the presence of unmanaged cat populations in public spaces. This approach is problematic in the Australian context, where it is strongly opposed by conservation groups, could encourage abandonment of owned cats, and may imply that some cats are entitled to a lower standard of care. Instead, efforts are better directed at reducing financial barriers, clarifying the legal meaning of “ownership,” and supporting semi-owners to formalise their caregiving role, while also addressing local sources of recruitment to the unowned cat population such as abandonment and migration.

The initial social science research within the KCSAH project identified six unique cat owner profiles amongst those who allowed their cats to roam [40]. These showed close alignment with the typology previously reported by Crowley et al. [48], suggesting consistent behavioural patterns across different contexts. More recently, Chamberlain et al. [49] used latent profile analysis to categorise cat owners into four distinct COM-B segments: *engaged* (6%), *receptive* (17%), *ambivalent* (48%), and *opposed* (30%). Despite methodological differences, these studies converge in highlighting that a large proportion of cat owners fall into the middle ground of ambivalence or limited capability. At the same time, a smaller subset is either highly motivated to contain cats or strongly resistant to change.

A key limitation of the KCSAH project was the inability to robustly assess the effects of targeted messaging on human behaviour change. While the campaign was grounded in strong formative research and designed around owner segments, the primary evaluation relied on pre- and post-intervention surveys that used different methodologies. This limited the capacity to directly attribute changes in containment behaviour to the campaign. The post-intervention panel survey revealed that cat owners who recalled exposure to campaign messaging reported higher COM-B scores, indicating improved capability, opportunity, and motivation for containment. However, because this assessment occurred late in the project, it remains unclear how these attitudinal shifts subsequently translated into sustained human behaviour change. Importantly, stated attitudes and intentions do not always align with real-world practices, as illustrated in the willingness-to-pay literature, where expressed support for higher-welfare options often exceeds actual purchasing behaviour [50,51].

Similar challenges were encountered in assessing the impact and return on investment of the different channels used to raise awareness of the project. For example, the social media marketing campaign, which accounted for 17% of the project budget, achieved the greatest reach, with over 3.5 million social media impressions and more than 42,000 blog/article views. By contrast, the school education programme represented only 1.2% of the budget and directly reached 1404 students, while the printed/packaged resources (4.8% of the budget) reached just over 14,000 caregivers. Although these latter channels had a much higher per-person cost and lower reach than mass digital strategies, they may have been more effective in influencing behaviour change by fostering deeper engagement and longer-term shifts in norms. Compared to mass communication channels, face-to-face discussions—particularly with trusted advisors such as veterinarians—offer opportunities to address caregiver concerns directly, provide practical and context-specific solutions, and reinforce social norms surrounding cat care [39,52]. However, more robust data are needed on how effectively veterinarians and rescue organisations engaged in these conversations after receiving partner packages, as little is known about whether the materials altered their approach, how frequently they raised the topic with caregivers, or the extent to which their own views shaped the quality and impact of those discussions.

For veterinarians in particular, workforce capacity and lack of training and confidence in desexing unsocialized or prepubertal cats can significantly limit their ability to engage in community cat programmes, as evident in the implementation of the KCSAH targeted desexing initiative. A total of 2731 cats were desexed through 26 private veterinary clinics that participated in the KCSAH project at a programme cost of AUD 829,861, equating to an average of approximately AUD 304 per cat. By contrast, Cotterell et al. [53] reported substantially lower costs in a Victorian targeted desexing programme, which sterilised 831 cats between 2013 and 2021 for a total cost of AUD 77,490, or about AUD 93 per cat, including microchipping. It should be noted, however, that such low costs are only achievable where private veterinarians heavily subsidise their fees. The higher average cost reported in KCSAH is therefore more likely to reflect the true cost of delivering desexing at scale. Importantly, veterinarians should not be expected to absorb these costs; the real costs of service delivery need to be recognised and built into programme planning and funding. As desexing is a critical part of controlling cat populations, veterinary sector capacity building in high-quality, high-volume spay–neuter techniques, anaesthesia, and pain management for unsocialized or prepubertal cats would be beneficial, especially by incorporating this into veterinary curricula and leveraging partnerships with veterinary schools [54,55,56]. Long-term success will also depend on sustained funding and cross-council coordination, particularly in regions where cat migration between jurisdictions undermines local efforts.

Whether driven by desexing efforts or human behaviour change, monitoring data indicated that the KCSAH project contributed to positive ecological and community outcomes in several participating councils, although results were uneven across sites. Ecological surveys suggested reductions in free-roaming cat densities in participating councils, while camera data showed persistent cat roaming activity in the same areas, indicating that while cat population density decreased, they continued to occupy the same spaces. Community-level indicators were more consistent, with many councils experiencing notable declines in nuisance complaints and cat impoundments. In councils that reported increases, this was likely due to either the more limited scope of KCSAH activities or to greater community awareness prompting more proactive reporting of free-roaming cats rather than an actual rise in nuisance activity. Overall, these findings are consistent with international evidence indicating that accessible, community-focused desexing initiatives can lead to measurable reductions in intake, euthanasia, and nuisance complaints when implemented at scale and sustained over time. Examples include Banyule City Council’s low-cost desexing programme, which delivered a 469% return on investment by reducing animal management expenses [53]; the Rosewood Community Cat Program in rural Australia, which achieved declines of up to 85% in euthanasia and 60% in intake within three years [57]; and U.S. initiatives such as the New Hampshire Animal Population Control Program, which linked subsidised sterilisation to long-term reductions in shelter admissions [58]. Together, these examples reinforce that targeted, well-supported desexing programmes can underpin ecological and community benefits, particularly when combined with behaviour change and broader welfare interventions.

Overall, the KCSAH project underscores the need for more systematic and standardised approaches to evaluating cat management interventions. Readily available metrics such as social media analytics, website traffic, and resource downloads can demonstrate reach and engagement, but they do not directly measure human behaviour change. Survey-based social science research provides deeper insights into motivations and barriers, yet it is costly to conduct and is increasingly affected by declining response rates [59,60,61]. Ecological monitoring methods, such as camera trapping, provide robust evidence of cat activity but are resource-intensive and unlikely to be sustainable for most jurisdictions. Other commonly used metrics, such as the number of registered cats and cat-related nuisance complaints, proved to be of limited value, as both are strongly influenced by reporting effort and compliance processes rather than reflecting actual changes in cat populations or their impacts. More practical, scalable indicators may include shelter and pound intake data. However, these too require cautious interpretation, as increases may reflect improved reporting or changing policies, rather than worsening conditions. Developing a nationally consistent monitoring and reporting framework would greatly strengthen the ability to compare interventions across regions and track progress over time.

Finally, the sustainability of large-scale human behaviour change programmes is a critical consideration. The KCSAH project was supported by a sizable grant from the NSW Environment Trust; however, actual expenditure was likely underestimated, as the project was able to leverage existing RSPCA infrastructure, staff expertise, and communication platforms. Some costs were one-off investments in resource development, but maintaining engagement over time requires dedicated full-time staff positions. Although ownership of the project was transferred to RSPCA NSW, many initiatives risk losing momentum once initial funding ends, as they are labour-intensive to sustain. Ongoing investment is therefore essential to support staff roles at state or regional levels to coordinate desexing and outreach across multiple councils. Shared or co-funded positions could reduce costs while ensuring continuity, and opportunities exist to centralise and coordinate promotion of low-cost desexing initiatives, an approach already adopted by some organisations in Australia, to minimise duplication and increase reach.

## 5. Conclusions

The KCSAH project showed that large-scale behaviour change programmes can engage communities, reduce free-roaming cat impacts, and strengthen cross-sector partnerships. The project reached millions through social marketing and digital resources, while thousands of people engaged directly via community events, school programmes, and subsidised desexing. Ecological monitoring revealed reductions in roaming cat densities in several councils, with many areas also experiencing declines in nuisance complaints and impoundments. However, results were uneven, and survey data could not robustly demonstrate how increased capability, opportunity, and motivation translated into sustained containment behaviours. These limitations underscore the importance of establishing consistent evaluation frameworks and investing in metrics that are practical for councils to collect and interpret.

At the same time, the project exposed structural challenges. Workforce capacity, funding, and practitioner confidence limit the ability of private veterinary practices to deliver high-volume, low-cost desexing. The actual costs of service delivery must be built into programme planning, rather than relying on veterinary subsidies. Legal definitions of ownership, registration requirements, and the reluctance of some caregivers to assume full responsibility for semi-owned cats also remain barriers. Addressing these issues will require ongoing policy reform, expanded veterinary training, and resource models that support councils and communities in working together across jurisdictions. By embedding behaviour change science within integrated programmes that combine education, desexing, and supportive policy, the foundations established by KCSAH can inform future national strategies to improve cat welfare, reduce conflict with communities, and protect Australia’s biodiversity.

## Figures and Tables

**Figure 1 animals-15-03554-f001:**
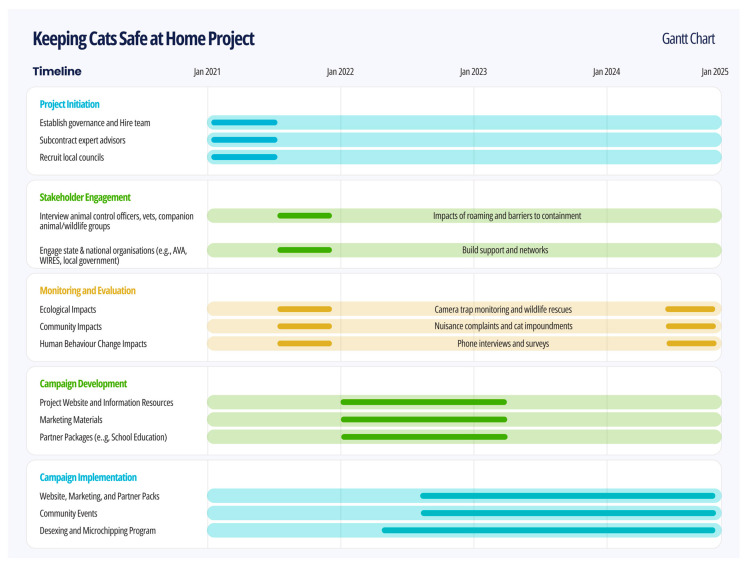
Timeline of the staged implementation of the Keeping Cats Safe at Home (KCSAH) project.

**Figure 2 animals-15-03554-f002:**
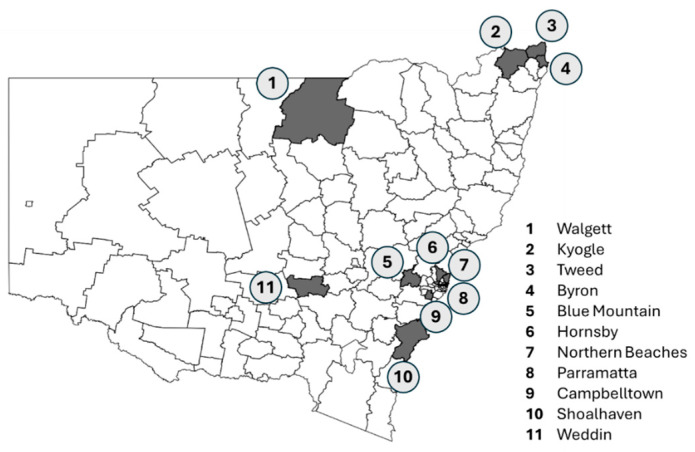
Geographic location of 11 councils in New South Wales, Australia, participating in the KCSAH project.

**Figure 3 animals-15-03554-f003:**
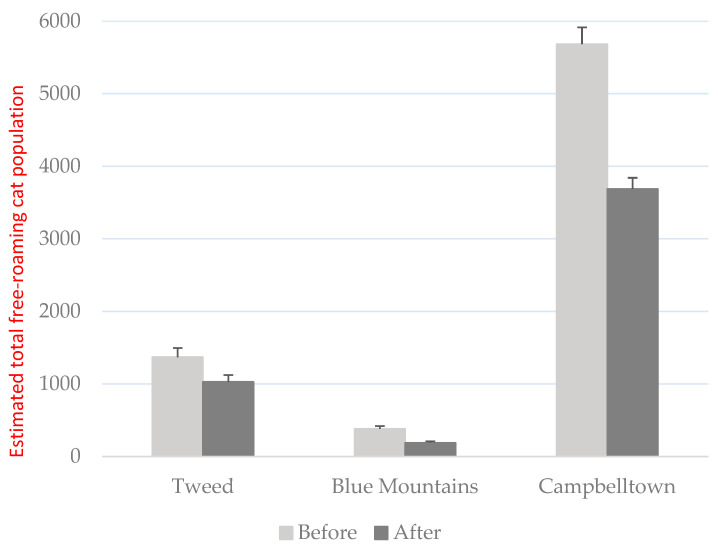
Free-roaming cat population estimates before and after the Keeping Cats Safe at Home project in the Tweed, Blue Mountains, and Campbelltown local government areas.

**Table 1 animals-15-03554-t001:** Demographic and socioeconomic characteristics of the 11 councils in New South Wales, Australia, participating in the KCSAH project.

Council	Population Size (Number)	Land Area (Km^2^)	Population Density (Residents Per Km^2^)	Unemployment Rate (%)	Median Weekly Household Income (AUD)	Speaks Language Other than English at Home (%)	Socio-Economic Index Rank (1 Low, 128 High)
Blue Mountains	77,913	1431	55	2.5	AUD 1756	6.5	110
Byron	36,510	566	66	2.4	AUD 1602	9.6	100
Campbelltown	180,365	311	593	4.7	AUD 1700	35.8	31
Hornsby	151,747	455	339	3.8	AUD 2417	35.7	120
Kyogle	9453	3584	3	4.1	AUD 983	3.3	13
Northern Beaches	263,298	254	1054	2.1	AUD 2592	15.6	121
Parramatta	260,379	84	3211	2.3	AUD 2051	56.4	104
Shoalhaven	109,611	4567	24	2.1	AUD 1250	49	65
Tweed	97,969	1308	76	2.3	AUD 1296	5	78
Walgett	5516	22,308	<1	9.5	AUD 1001	5.3	5
Weddin	3614	3415	1	1.8	AUD 1046	1.5	41

Data obtained from the Australian Bureau of Statistics 2021 Census and NSW Office of Local Government.

**Table 2 animals-15-03554-t002:** Summary of KCSAH expenditures by activity and financial year over the project lifespan.

Project Activity	Year 1 2020 to 2021	Year 2 2021 to 2022	Year 3 2022 to 2023	Year 4 2023 to 2024	Total	% of Total Budget
Salaries and Consultancy	AUD 76,625	AUD 253,049	AUD 188,597	AUD 438,867	AUD 957,138	38.0%
Project Administration	AUD 239	AUD 6300	AUD 2000	AUD 12,300	AUD 20,839	0.8%
Ecological Monitoring	-	AUD 35,332	AUD 2446	AUD 7913	AUD 45,691	1.8%
Website and Information Resources	AUD 0	AUD 8411	AUD 57,198	AUD 55,319	AUD 120,928	4.8%
Social Media Marketing Campaign	-	AUD 21,379	AUD 73,885	AUD 340,877	AUD 436,140	17.3%
Partner Packages	-	AUD 224	AUD 11,287	AUD 41,355	AUD 52,865	2.1%
School Education Package	-	-	AUD 21,844	AUD 7931	AUD 29,774	1.2%
Community Events and Engagement	AUD 483	AUD 2855	AUD 5984	AUD 18,879	AUD 28,201	1.1%
Targeted Desexing Programme	-	AUD 28,412	AUD 335,106	AUD 466,343	AUD 829,861	32.9%
**Total**	**AUD 77,347**	**AUD 355,962**	**AUD 698,345**	**AUD 1,389,783**	**AUD 2,521,437**	**100%**

**Table 3 animals-15-03554-t003:** Summary of KCSAH project reach across digital, community, education, and professional channels.

Outcome	Target	Result	Notes
Project webpage views	20,000	44,666	70% of visits from NSW; majority from Greater Sydney.
Blog/article views	-	42,474	Across 22 total blog articles
Resource downloads	5000	15,920	Includes resources (2566) and blogs (13,354).
Reach from social media posts (number of likes/impressions)	3,000,000	3,530,352	Across multiple campaign waves from 2022 to 2024.
Cat caregivers directly participating in desexing and microchipping	1000	4061	-
Community event attendees	20,000	36,858	Across more than 30 events
Students reached (school incursions)	1000	1404	-
Cat caregivers directly receiving digital or printed resources	10,000	14,233	Includes adopters (8234), desexing participants (1570), and newsletter subscribers (4429).
Veterinary practices engaged	10 (subsidised services)/20 (supporting)	26/51	100% of local veterinarians engaged in some form.
Other local organisations engaged	10	31	Includes councils, rehoming groups, community organisations.

**Table 4 animals-15-03554-t004:** Cat density estimates (cats per hectare) for each local government area calculated using line transect distance sampling.

LGA	Density	se	cv	95% CI	df
**Before**					
Tweed	0.24	0.02	0.09	0.20–0.29	62.41
Blue Mountains	0.24	0.03	0.11	0.19–0.30	16.46
Campbelltown	0.57	0.02	0.04	0.52–0.62	71.82
**After**					
Tweed	0.18	0.02	0.09	0.15–0.22	48.89
Blue Mountains	0.12	0.01	0.12	0.09–0.16	6.63
Campbelltown	0.37	0.02	0.04	0.34–0.40	38.13

LGA = local government area; Density = number of cats per hectare; se = standard error; cv = coefficient of variation; 95% CI = 95% confidence interval; df = degrees of freedom.

**Table 5 animals-15-03554-t005:** Summary of community outcomes from KCSAH monitoring.

Outcome	Target	Result	Notes
Reduction in nuisance cat complaints to councils	20% reduction	Reductions in Campbelltown (−64%), Parramatta (−51%), Hornsby (−47%), Northern Beaches (−61%); increases in Blue Mountains (+78%), Byron (+36%)	Complaint data collection varied between councils.
Reduction in cats impounded	10% reduction	Major reductions in Blue Mountains (−54%), Campbelltown (−59%), Parramatta (−73%), Walgett (−100%); increases in Northern Beaches (+53%), Weddin (+380%)	Increase in Weddin reflects improved community engagement; Northern Beaches had no targeted desexing programme.

**Table 6 animals-15-03554-t006:** Summary of human behaviour change outcomes from KCSAH monitoring.

Outcome	Target	Result	Notes
Uptake of containment strategies by cat caregivers	60% (participants)/40% (LGA population)	Insufficient survey responses (<10%) for direct comparison; panel survey used instead	Cat caregivers exposed to containment messaging reported higher capability, opportunity, and motivation to contain cats.
Cats desexed through the project	1000	2731	Includes 877 pre-pubertal desexings (target 300).
New cats registered	1000	17,464	Proportional increases ranged from +2% (Shoalhaven) to +75% (Weddin).
Cats microchipped	1000	1721	-
Cat enclosures purchased with discount codes	300	31	Many owners used alternative containment methods (indoor-only, fencing, supervised access).

## Data Availability

The raw data supporting the conclusions of this article will be made available by the authors on request.

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
