# Peer review of "Keeping Cats Safe at Home (KCSAH): Lessons Learned from a Human Behaviour Change Campaign to Reduce the Impacts of Free-Roaming Domestic Cats"

_animals, 2025, doi:10.3390/ani15243554_

Round 1

Reviewer 1 Report

Comments and Suggestions for Authors

General comments

I appreciate the straightforward nature of this study. The manuscript is well structured and the writing clear, making it easy to follow. Especially important are the lessons learned. That said, I do think some minor revisions are in order, as described below.

Specific comments

Lines 66–70

“While several studies have shown that measures such as using collars fitted with anti-predation devices [4-7] or incorporating different feeding programs or structured play into domestic cat care routines [8, 9] can help to reduce predation rates, they do not completely eliminate hunting, roaming, and other perceived nuisance behaviors [10, 11]”

I think the study by Herrera et al. (2022) must have been included in error; I don’t see anything there having to do with “different feeding programs or structured play.”

Lines 96–100

“Rand et al. (2024) found that many owned cats were not desexed, had inadequate containment, or lacked routine veterinary care, increasing both individual welfare risks and the likelihood of contributing to unwanted litters. Similar patterns were reported by Dutton-Regester and Rand (2024) in households enrolling cats in a free sterilization program…”

If these previous studies provided specific rates (e.g., cats not desexed), please briefly provide those here.

Lines 122–24

“As a result, there has been an increasing reliance on the voluntary uptake of cat containment practices; yet, barriers remain, including cost, housing design, and cultural beliefs about roaming as a ‘natural’ behavior [31-33].”

The use of quotes around “natural” implies that roaming is not natural. If that’s your intention, I think it warrants a brief explanation as to why roaming is unnatural—and, presumably, containment is natural.

Figure 1

Each of the colors is shown with two levels of saturation—what’s the difference? (It seems like the lighter versions are unnecessary.)

Lines 385–88

“Additional ecological insights were drawn from a complementary research project examining cat movement patterns and wildlife interactions at Sydney’s North Head (Kennedy et al., 2024). Together, these datasets allowed assessment of changes in roaming cat density and potential shifts in local wildlife activity.”

For what it’s worth, I don’t see that Kennedy et al. reported any “wildlife interactions”—only photos of cats, foxes, and other animals. To be clear, I glanced through the paper only briefly—perhaps I missed something important.

Figure 3 and Table 4

Please add units of measure to all density figures.

Lines 471–74

“The rise [of impoundments] in Weddin was attributed to improved community engagement and trust in council staff, leading to greater reporting and handover of cats. In contrast, Northern Beaches did not implement targeted desexing programs during the project.

Also, I’m confused about the connection between “trust in council staff” and the increased impoundments (nearly four-fold). In Weddin. In my experience, residents are less likely to trust animal services agencies when cats are removed only to be euthanized. It sounds like the opposite occurred here. Please clarify.

I’m curious, too, about why no targeted desexing program was implemented in Northern Beaches. Previously, you noted: “free desexing and microchipping programs were established in each of the 11 KCSAH partner council areas, with more intensive ‘StrayCare’ initiatives implemented in six councils to improve uptake among semi-owned and unowned cat groups that contribute disproportionately to new litters” (Lines 346–49).

Lines 480–85

“The post-intervention survey included 2,039 cat owners from across NSW. Results showed that 46% of respondents reported 24-hour containment, 26% practiced night curfews, and 28% allowed their cats to roam freely. This represented a lower proportion of cats reported as fully contained compared with the baseline survey in 2021, where 65% of owners indicated 24-hour containment, 24% practiced a night curfew, and 11% allowed free roaming (Ma & McLeod, 2023).”

Just a comment: This suggests a 29% reduction in 24-hour containment over a four-year period; that seems rather implausible. It might be worth a brief note of explanation—even if it’s conjecture.

Lines 548–50

“Program success improved when participation requirements, such as mandatory microchipping, were minimized, and assurances were provided that enforcement would not follow.”

The previous comment that “trust in council staff [led] to greater reporting and handover of cats” seems to contradict this. Please clarify.

Lines 587–89

“For semi-owners, the requirement to register cats and pay the associated lifetime fee was a major deterrent to microchipping during the KCSAH targeted desexing campaign, even though the microchipping was free.”

Please specify the cost of this lifetime fee.

Lines 600–610

“Other countries, such as Italy, Greece, Spain, and Brazil, have adopted colony caretaker models that authorize community members to feed and monitor free-roaming cats without assuming full liability. While such models reduce barriers, they also risk normalizing the presence of unmanaged cat populations in public spaces. This approach is problematic in the Australian context, where it is strongly opposed by conservation groups, could encourage abandonment of owned cats, and may imply that some cats are entitled to a lower standard of care. Instead, efforts are better directed at reducing financial barriers, clarifying the legal meaning of “ownership,” and supporting semi-owners to formalize their caregiving role, while also addressing local sources of recruitment to the unowned cat population such as abandonment and migration.”

Please add some citations regarding programs in Italy, Greece, Spain, and Brazil.

Literature cited

Dutton-Regester, K., & Rand, J. (2024). Cat Caring Behaviors and Ownership Status of Residents Enrolling a Cat in a Free Sterilization Program. Animals, 14(20), 3022. https://doi.org/10.3390/ani14203022

Herrera, D. J., Cove, M. V., McShea, W. J., Flockhart, D. T., Decker, S., Moore, S. M., & Gallo, T. (2022). Prey selection and predation behavior of free-roaming domestic cats (Felis catus) in an urban ecosystem: Implications for urban cat management. Biological Conservation, 268, 109503. https://doi.org/10.1016/j.biocon.2022.109503

Kennedy, B. P. A., Clemann, A., & Ma, G. C. (2024). Feline Encounters Down Under: Investigating the Activity of Cats and Native Wildlife at Sydney’s North Head. Animals, 14(17), 2485. https://doi.org/10.3390/ani14172485

Kerr, A. C., Rand, J., Morton, M. J., Reid, R., & Paterson, M. (2018). Changes Associated with Improved Outcomes for Cats Entering RSPCA Queensland Shelters from 2011 to 2016. Animals, 8(6). https://doi.org/10.3390/ani8060095

Legge, S., Murphy, B. P., McGregor, H., Woinarski, J. C. Z., Augusteyn, J., Ballard, G., Baseler, M., Buckmaster, T., Dickman, C. R., Doherty, T., Edwards, G., Eyre, T., Fancourt, B. A., Ferguson, D., Forsyth, D. M., Geary, W. L., Gentle, M., Gillespie, G., Greenwood, L., … Zewe, F. (2017). Enumerating a continental-scale threat: How many feral cats are in Australia? Biological Conservation, 206(February), 293–303. https://doi.org/10.1016/j.biocon.2016.11.032

Ma, G. C., & McLeod, L. J. (2023). Understanding the Factors Influencing Cat Containment: Identifying Opportunities for Behaviour Change. Animals, 13(10), Article 10. https://doi.org/10.3390/ani13101630

Rand, J., Scotney, R., Enright, A., Hayward, A., Bennett, P., & Morton, J. (2024). Situational Analysis of Cat Ownership and Cat Caring Behaviors in a Community with High Shelter Admissions of Cats. Animals, 14(19), Article 19. https://doi.org/10.3390/ani14192849

Tan, K., Rand, J., & Morton, J. (2017). Trap-Neuter-Return Activities in Urban Stray Cat Colonies in Australia. Animals, 7(6), 46. https://doi.org/10.3390/ani7060046

Author Response

Please see file attached.

Reviewer 2 Report

Comments and Suggestions for Authors

This article presents a relevant and innovative experience in the management of domestic cats through human behavior change strategies. The Keeping Cats Safe at Home (KCSAH) project represents an original application of the COM-B model to the field of animal welfare and conservation, providing a valuable interdisciplinary perspective that combines social communication, education, and local policy development.

However, the scale of investment (AUD 2.5 million) contrasts with the limited evidence of measurable impact. Although the authors document improvements in owners’ attitudes and awareness, the assessment of actual behavioral change remains insufficient and relies largely on self-reported data. The absence of a control group, longitudinal design, and robust ecological indicators restricts the external validity of the conclusions.

Furthermore, the study lacks a cost-effectiveness analysis and a clear projection of institutional sustainability once funding ends. The ecological impact is inferred from indirect indicators (e.g., reduction in complaints, sightings, or shelter admissions) without quantitative evidence of benefits to wildlife.

Despite these limitations, the paper provides a conceptual framework that is potentially exportable and adaptable to other contexts—particularly its use of positive messaging, community education, and intersectoral collaboration. The work is valuable as a pilot experience and methodological reference, though its practical reproducibility would require a more resource-efficient design, stronger evaluation methods, and long-term sustainability planning.

In summary, this is a socially and conceptually relevant study, with modest demonstrated effectiveness and emerging empirical evaluation. Its publication contributes to the ongoing discussion on integrating behavioral science into cat welfare and management policies.

Reviewer 3 Report

Comments and Suggestions for Authors

Overall, the manuscript was clear and well written.  Being a case study, it may be slightly limited in its applicability to other municipalities.

I have a few minor points for the authors to consider:

Figure 3:  Labeling the Y-axis of the graph (e.g., Cats / km²) would improve its readability.  Minimally, put the precise information in the figure caption (e.g., is it cats / km², cats / mile², cats / acre?).

Figure 4 caption:  A single number cannot be an interval "lcl = lower 95% confidence interval".  Do you mean "lcl = lower limit of the 95% confidence interval"?  Likewise for "ucl".  Consider replacing the lcl and ucl columns with a single column labeled 95% CI and putting the lower and upper limits as a range in that column (e.g. 0.20 - 0.29).

Figure 4:  Should the UCL for Tweed before the intervention be 0.29 instead of .029?

My personal preference is to not split tables across pages.  If you do split tables across pages, you should repeat the row with column headings on each page.

Line 469-471:  To be consistent, report the percent increases for Northern Beaches and Weddin as you report the percent decreases for the other communities.
